

# ALL-Net: integrating CNN and explainable-AI for enhanced diagnosis and interpretation of acute lymphoblastic leukemia

Abhiram Thiriveedhi[1,2], Swetha Ghanta[1], Sujit Biswas[3,4] and Ashok K. Pradhan[1]

[1] Department of Computer Science and Engineering, School of Engineering and Sciences, SRM University, AP, Guntur, Andhra Pradesh, India
[2] Department of Computer Science, State University of New York at Stony Brook, New York, United States
[3] Computer Science, University of London, London, United Kingdom
[4] Computer Science Department, University of Northumbria at Newcastle, Newcastle, United Kingdom

Corresponding authors
Sujit Biswas,
sujit.biswas@northumbria.ac.uk
Ashok K. Pradhan,
ashokkumar.p@srmap.edu.in

## ABSTRACT

This article presents a new model, ALL-Net, for the detection of acute lymphoblastic leukemia (ALL) using a custom convolutional neural network (CNN) architecture and explainable Artificial Intelligence (XAI). A dataset consisting of 3,256 peripheral blood smear (PBS) images belonging to four classes—benign (hematogones), and the other three Early B, Pre-B, and Pro-B, which are subtypes of ALL, are utilized for training and evaluation. The ALL-Net CNN is initially designed and trained on the PBS image dataset, achieving an impressive test accuracy of 97.85%. However, data augmentation techniques are applied to augment the benign class and address the class imbalance challenge. The augmented dataset is then used to retrain the ALL-Net, resulting in a notable improvement in test accuracy, reaching 99.32%. Along with accuracy, we have considered other evaluation metrics and the results illustrate the potential of ALLNet with an average precision of 99.35%, recall of 99.33%, and F1 score of 99.58%. Additionally, XAI techniques, specifically the Local Interpretable Model-Agnostic Explanations (LIME) algorithm is employed to interpret the model's predictions, providing insights into the decision-making process of our ALL-Net CNN. These findings highlight the effectiveness of CNNs in accurately detecting ALL from PBS images and emphasize the importance of addressing data imbalance issues through appropriate preprocessing techniques at the same time demonstrating the usage of XAI in solving the black box approach of the deep learning models. The proposed ALL-Net outperformed EfficientNet, MobileNetV3, VGG-19, Xception, InceptionV3, ResNet50V2, VGG-16, and NASNetLarge except for DenseNet201 with a slight variation of 0.5%. Nevertheless, our ALL-Net model is much less complex than DenseNet201, allowing it to provide faster results. This highlights the need for a more customized and streamlined model, such as ALL-Net, specifically designed for ALL classification. The entire source code of our proposed CNN is publicly available at https://github.com/Abhiram014/ALL-Net-Detection-of-ALL-using-CNN-and-XAI.

## INTRODUCTION

Leukemia is a group of cancers that starts in the bone marrow and affects the body's blood-forming tissues. It is caused by the unregulated buildup of atypical white blood cells, which interferes with normal blood cell operations and results in different types of complications. Frequent signs of leukemia include chills or fever, recurring or serious infections, reducing weight naturally, swelling of the lymph nodes, spleen or liver enlargement, frequent nosebleeds, petechiae, or little red dots on skin, persistent perspiration, particularly at night, bone sensitivity or pain (*Mayo Clinic, 2025*). In 2021, according to data from the Surveillance, Epidemiology, and End Results (SEER) database, there were approximately 61,090 projected new cases of leukemia, constituting 3.2% of all newly diagnosed cancer cases. This places leukemia as the tenth most prevalent cancer in the United States (*Chennamadhavuni et al., 2023*).

Acute lymphoblastic leukemia (ALL), a type of leukemia, is a fast-moving leukemia that targets lymphoid cells, a subset of white blood cells that are part of the immune system. This disease occurs mostly in children, but sometimes adults are also prone to it. There are several ways to detect ALL (*City of Hope, 2024*). Some of them are:

- **Blood tests**: These tests comprise of peripheral blood smear (PBS) examination and a complete blood count (CBC) to detect abnormalities in blood cell counts and morphology.
- **Bone marrow biopsy**: A sample of bone marrow is obtained to confirm the diagnosis and assess bone marrow involvement.
- **Immunophenotyping**: This technique analyzes surface proteins on leukemia cells to differentiate between subtypes and confirm the diagnosis.
- **Cytogenetic and molecular testing**: These tests identify genetic abnormalities associated with ALL, aiding in risk stratification and treatment planning.
- **Lumbar puncture**: Sometimes performed to assess leukemia spread to the central nervous system.
- **Imaging studies**: X-rays, ultrasound, CT, or MRI may evaluate disease involvement in other organs.

Detecting ALL poses challenges because of the limitations of current diagnostic methods. Invasive procedures like bone marrow aspiration and biopsy carry risks and can yield inconclusive results. Bone marrow biopsy is largely expensive and can generate great pain to the patients. It could not be suitable with some patients particularly children, in whom the disease is more prevalent. Immunophenotyping and genetic testing provide valuable insights but may not always distinguish between closely related subtypes or detect all genetic abnormalities (*Rezayi et al., 2021*). Imaging studies and clinical assessments, while informative, may miss early or atypical presentations of the disease. Overall, a

multidisciplinary approach is essential to navigate the complexities of ALL diagnoses and ensure accurate and timely identification of the disease. While current techniques such as blood tests, bone marrow analysis, and imaging studies play vital roles in diagnosis, they are prone to subjective interpretation, interobserver variability, and potential errors. For blood tests and PBS images, the major disadvantage is that they require skilled experts to manually examine microscopic images of blood or bone marrow samples, leading to significant delays in the treatment process. These tests are also prone to error (*Gehlot, Gupta & Gupta, 2020*). Moreover, these elaborate diagnostic approaches are typically not employed for cases presenting with routine symptoms. Also, many diseases involve the abrupt rise of lymphocytes which should not be confused with leukemia (*McGrath, 2002*). These challenges call for an accurate, low cost and time-efficient, automated system of diagnosis. The disease must be detected as early and fast as possible thereby providing the patient more time for recovery and survival. The need for computer-aided diagnosis (CAD) in detecting ALL arises from the complexities and challenges associated with traditional diagnostic methods. CAD systems offer the potential to augment diagnostic accuracy by leveraging advanced algorithms to analyze peripheral blood smear images, immunophenotyping data, genetic profiles, and clinical information. Additionally, CAD systems can handle large volumes of data proving to be time-efficient.

This study mainly focuses on distinguishing ALL and its subtypes—Early-B, Pre-B, Pro-B from hematogones which are quite similar to ALL, using machine intelligence. In this article, we use the terms Early-B as Early, Pre-B as Pre, and Pro-B as Pro interchangeably. Hematogones also involve an abrupt rise of lymphocytes and mostly occur in infants and young children. Unlike ALL, hematogones are not harmful and they often go away as the children age. This visual similarity presents a challenge for accurate diagnosis, as both ALL and hematogones exhibit overlapping features in PBS images. However, distinguishing these cases is critical: hematogones are benign or harmless, whereas ALL requires immediate treatment. In this work, we utilized the PBS image dataset sourced from Kaggle (*Aria et al., 2021*). The PBS images of different subcategories of ALL can be seen in Fig. 1. Differentiating benign cells from Early-B cells, as well as Pre-B from Pro-B cells, can be difficult due to visual similarities. Therefore, the morphological characteristics of each stage, focusing on specific features like chromatin pattern, nucleoli visibility, and cytoplasmic granularity should be considered. This complexity necessitates the use of deep neural networks (DNNs), which can detect intricate patterns and classify images accordingly (*Bodzas, Kodytek & Zidek, 2020*).

The images underwent preprocessing including resizing and normalization. Then, to mitigate data imbalance, we applied data augmentation specifically to the benign class. Subsequently, the pre-processed images are inputted into our customized convolutional neural network (CNN), ALL-Net. This model is designed to identify complex features within the images and classify them into categories: benign (hematogones) and malignant subtypes of acute lymphoblastic leukemia (ALL) such as Early-B, Pre-B, and Pro-B. Finally, we evaluated the proposed CNN model on both augmented and unaugmented image datasets and rigorously evaluated the proposed model performance using various evaluation metrics. Additionally, to understand the mechanism behind the decision-

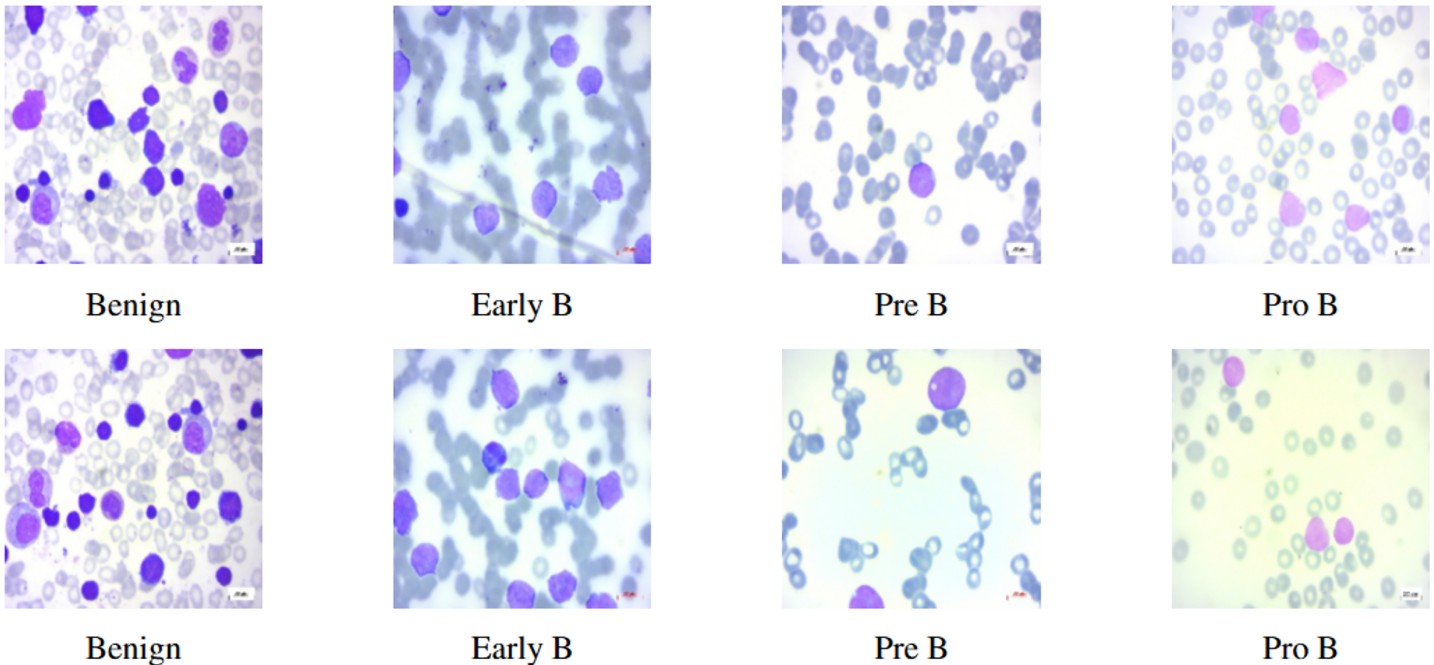

**Figure 1 Benign and ALL sub-types.**

making process of the CNN, we employed explainable Artificial Intelligence (XAI) algorithms, specifically the Local Interpretable Model-Agnostic Explanations (LIME) algorithm. This algorithm highlights the most significant regions in the image that influenced the CNN model's classification into its respective class. XAI is crucial in medical diagnosis because it helps doctors understand how AI makes decisions, thereby building trust in its use. XAI makes sure that the decisions are accurate and free from biases. This transparency also helps find mistakes and makes it easier to explain medical decisions to patients and regulators improving healthcare overall.

The main contributions of this work are:

1. We proposed a customized CNN model, ALL-Net, which is used to divide the PBS images into classes of benign, which represents the hematogones, and the malignant ALL subtypes—Early-B, Pre-B, Pro-B.
2. We used the PBS image dataset from Kaggle, and preprocessed them by resizing and normalizing. Then, we performed the data augmentation on the images of the benign class to solve the data imbalance problem.
3. To address the black-box nature of the ALL-Net model, we employed XAI, specifically the LIME algorithm, to enhance the interpretability of the results.

This article is structured to first explore existing research on leukemia detection, followed by an examination of foundational concepts relevant to this study. It then delves into a detailed description of the technique and application of the proposed ALL-Net

model. Subsequently, it presents the results generated by the ALL-Net model and their interpretations. Finally, the article concludes by discussing future research directions.

## LITERATURE REVIEW

Several studies have investigated the use of deep learning models—mostly CNN-based architectures—for the automatic categorization and identification of leukemia and ALL from various forms of data ranging from symptoms of the patient, and blood test reports to images of blood smears in the last few years. Numerous publicly accessible datasets, including the American Society of Hematology (ASH) Image Bank, ALL-IDB Acute Lymphoblastic Leukemia Image Database for Image Processing (ALL-IDB) (*Labati, Piuri & Scotti, 2011*), ALL Challenge, and locally gathered datasets from the IEEE International Symposium on Biomedical Imaging (ISBI) 2019, have been used in the investigations.

One of the earliest studies in this domain was conducted by *Arivuselvam & Sudha (2022)*, who employed a Deep CNN (DCNN) classifier along with traditional machine learning models like support vector machines, decision trees, naive bayes, and random forests. They achieved impressive accuracies of 99.2% and 98.4% on the ASH Image Bank and ALL-IDB datasets, respectively, using the DCNN model. However, their study is limited to a binary classification of ALL *vs.* non-ALL cases. Subsequently, *Al-Bashir, Khnouf & Bany Issa (2024)* compared the performance of different CNN-based algorithms, including AlexNet, DenseNet, ResNet, and VGG16, on the same datasets. They reported an accuracy of 94%, showcasing the potential of these pre-trained models for ALL classification. *Rahman et al. (2023)* considered a more comprehensive approach by addressing multiclass blood cancer classification using deep CNN models like VGG19, ResNet50, InceptionV3, and Xception, along with traditional machine learning models. They achieved an impressive accuracy of 99.84% on the ALL-IDB1 and ALL-IDB2 datasets, demonstrating the superiority of deep learning techniques for this task. *Shafique & Tehsin (2018)* explored the use of ensemble approaches, combining pre-trained CNN models such as ResNet50, VGG16, and InceptionV3. They reported an accuracy of 99.8% on the ALL_IDB1 and ALL_IDB2 datasets using an ensemble of these models, highlighting the potential of ensemble techniques for improving performance. *Sampathila et al. (2022)* proposed a customized deep learning classifier called ALLNET, specifically designed for ALL detection using the ALL Challenge dataset from ISBI 2019. With a 95.54% accuracy rate, their model proved the usefulness of customized architectures for this kind of work. While most studies focused on binary or multiclass classification, *He et al. (2020)* took a different approach by using CNN models to predict leukemia-related transcription factor binding sites from DNA sequence data, achieving an accuracy of 75%.

Some works were also executed in the field of medical image analysis using XAI. *Van der Velden et al. (2022)* provide a comprehensive review of the current state and developments in XAI techniques applied to medical image analysis. The survey categorizes various XAI approaches and discusses their applications across different medical imaging tasks such as chest, skin, kidney, cardiovascular, eye, *etc*. Special emphasis is placed on the adaptation of computer vision-based XAI methods to medical imaging, highlighting the inclusion of domain-specific knowledge. Regarding the works done on the diagnosis of

leukemia using medical data and XAI, the authors of *Islam, Assaduzzaman & Hasan (2024)* proposed an explainable supervised machine learning model designed to accurately predict early-stage leukemia based solely on symptoms. A survey conducted with both leukemia and non-leukemia patients identified sixteen essential features. The proposed model, centered around a decision tree classifier, outperforms other algorithms by generating transparent rules directly applicable in clinical settings. Employing the *apriori* algorithm for generating these rules, the study conducts feature analysis and selection to underscore individual feature strengths and enhance model performance. The model demonstrated an impressive perfomance, achieving 97.45% accuracy, a Matthew's correlation coefficient of 0.63, and an area under the receiver operating characteristic curve (ROC) of 0.783 on the test dataset. However, the data used in their study is tabular, unlike the image-based data utilized in our research.

It is worth observing that the above mentioned works varied in terms of dataset size, ranging from a few hundred to thousands of images, which could potentially impact the generalizability of the results. Additionally, some studies utilized locally collected datasets, which may introduce biases or variations in image quality and labeling. Eventually, the analysis of the literature shows how much progress has been made in using deep learning methods—in particular, CNN-based models—and XAI for medical image analysis. Through this work, our objective is to develop a system that leverages interpretable AI methods to enhance the interpretability of CNN-based image analysis. This approach not only addresses issues such as dataset biases and scarce data availability but also aims to utilize a large, diverse image dataset to improve the generalizability of the CNN model for diagnostic purposes.

## PRELIMINARY CONCEPTS

In this section, a brief introduction to CNN, data augmentation, and XAI, the concepts that are used in this article is given.

### Convolutional neural networks

A CNN (*O'shea & Nash, 2015*) is a deep learning algorithm specifically made for image processing and analysis. CNNs draw inspiration from the structure of the visual cortex in the human brain and have brought significant changes in fields like computer vision and image recognition. CNNs are made up of several layers, each of which has a distinct function throughout the picture analysis process. Here's a brief overview of the main types of layers in a typical CNN as follows:

1. **Convolutional layer**: The basic component of a CNN is the convolutional layer. The input image is subjected to convolution operations with the aid of filters, commonly referred to as kernels. These filters or kernels, move across the input image to pick up characteristics like edges and patterns. The output of this layer is a feature map that includes all of the image's significant features.

2. **Activation layer (ReLU)**: To add non-linearity to the network, a non-linear activation function is added following each convolutional operation. In order to aid the network in

learning intricate patterns and features, the Rectified Linear Unit (ReLU) activation function (*Agarap, 2018*) substitutes zero for negative pixel values.

3. **Pooling layer**: The feature maps produced by the convolutional layers are made smaller in size by the pooling layer. It uses procedures like max pooling and average pooling to combine data from nearby pixels. This layer contributes to reducing the network's computational complexity.

4. **Fully connected layer (dense layer)**: Classification is performed by the fully connected layer, utilizing features extracted by preceding layers. The network can learn intricate correlations among these features and generate predictions based on the input data due to its connectivity, linking each neuron in one layer with every neuron in the layer above. In image classification tasks, softmax activation units (*Kouretas & Paliouras, 2019*) in the output layer produce probability scores for each class.

Typically, these layers are placed one after the other to construct the CNN's architecture. As data moves through these layers, the network gains the ability to extract progressively more abstract properties from the input image, which eventually helps it to make precise predictions or classifications. In summary, CNNs are effective tools for applications like object detection, image recognition, and medical image analysis because they automatically learn information about the image through the use of convolutional, activation, pooling, and fully connected layers.

## DATA AUGMENTATION

Data augmentation is a technique used in machine learning and deep learning, particularly in tasks involving images, where the original dataset is enhanced by applying various changes to the images, such as rotation, translation, flipping, and color adjustments. By creating these modified versions of the original images, the training dataset effectively expands, which helps the model learn to generalize better to different variations and conditions it might encounter during deployment. This technique is crucial for improving model performance and robustness, especially when the available training data is limited or when dealing with diverse real-world scenarios. In data augmentation for image processing, several operations can be performed to generate variations of the original images. Some common operations can be seen in Table 1. The process of data augmentation can be understood visually in Fig. 2. The input image, displayed on the left side of the figure, is a benign PBS image. The data augmentation techniques are applied to enhance the CNN model's ability to learn and generalize from the image.

### Explainable AI

XAI refers to a set of algorithms and methods developed to ensure that the workings of artificial intelligence (AI) systems can be understood by humans. The primary goal of XAI is to make AI systems more transparent and their decision-making processes more accessible and comprehensible to users, stakeholders, and regulators. This is particularly important in applications where AI decisions have significant impacts, such as in healthcare, finance, and criminal justice. XAI techniques are broadly classified into two

**Table 1 Data augmentation operations.**

| Operation | Description |
|---|---|
| Rotation | Rotating the image by a certain angle |
| Translation | Shifting the image horizontally or vertically |
| Scaling | Resizing the image while maintaining its aspect ratio |
| Shearing | Tilting the image along a certain axis |
| Flipping | Mirroring the image horizontally or vertically |
| Color jittering | Adjusting contrast, brightness, saturation |
| Noise injection | Adding random noise to the image |
| Crop and pad | Cropping or padding the image to a specific size |
| Elastic deformation | Distorting the image using elastic transformations |
| Random erasing | Randomly erasing parts of the image |

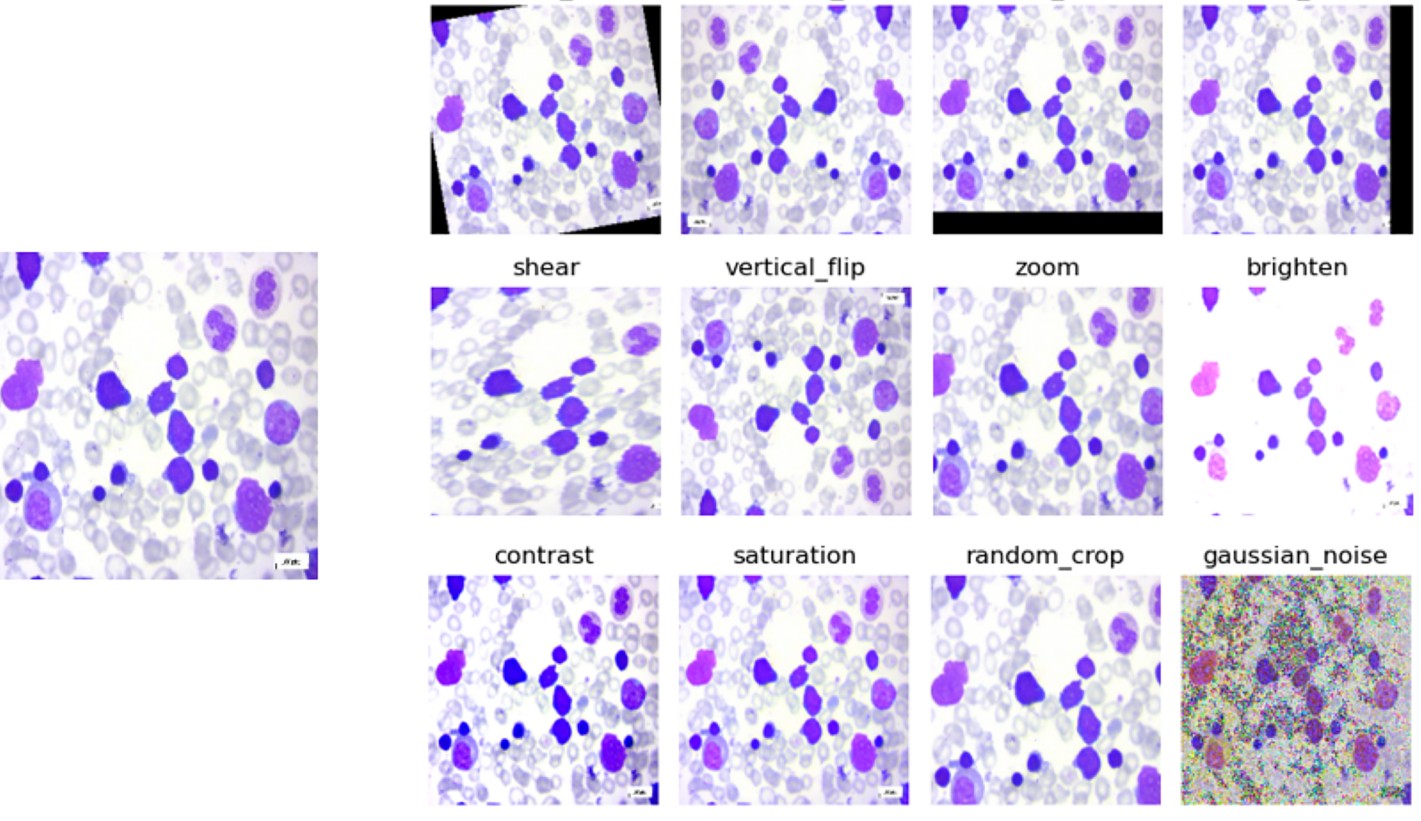

**Figure 2 Data augmentation example on a benign class image.**

categories: model-agnostic and model-specific. Model-agnostic techniques can be deployed on any ML or CNN model regardless of their internal structure. Some of these include LIME (Local Interpretable Model-agnostic Explanations) and SHAP (SHapley

Additive exPlanations) (*Lundberg & Lee, 2017*). Model-specific techniques are designed for specific types of models and leverage their unique characteristics for explainability. These include decision trees and Grad-CAM (*Selvaraju et al., 2016*). In this work, we used LIME to interpret the results generated by the proposed ALL-Net model.

### LIME

LIME explains individual predictions made by complex image classification models by approximating the model's behavior locally (*Ribeiro, Singh & Guestrin, 2018*). For image classification, LIME first segments the image into superpixels, which are contiguous regions with similar characteristics. It then generates a set of perturbed images by randomly altering these superpixels, such as turning some regions on and off. The original model is queried to obtain prediction probabilities for each perturbed image. These perturbed instances are weighted based on their similarity to the original image. Finally, LIME trains an interpretable surrogate model, such as a linear model, on these weighted instances to approximate the original model's decision-making process. The coefficients of the surrogate model reveal the importance of each superpixel, providing a visual and understandable explanation of the original model's prediction.

Mathematically, LIME approximates the local behavior of the CNN by fitting a linear surrogate model to these perturbed instances. Let $f$ be the original complex model, and $x$ be the input image. For each perturbed version $x'$ of the input $x$, LIME computes the prediction $f(x')$ and assigns a weight $\pi(x, x')$ that measures the similarity between the original and the perturbed instance. The goal is to solve:

$$argmin_{g \in G} \sum_{x'} \pi(x, x')(f(x') - g(x'))^2 + \omega(g) \tag{1}$$

where $g$ is the linear surrogate model, $G$ represents the class of interpretable models, and $\omega(g)$ is a regularization term for ensuring simplicity in $g$. The surrogate model $g$ approximates the decision boundary of the original CNN model in the local neighbourhood of $x$. The coefficients of $g$ reveal the importance of each superpixel, visually highlighting the most influential regions in the original image, offering an interpretable explanation for the model's prediction.

## METHODOLOGY

This section describes our proposed work which is arranged as follows: (a) Data collection. (b) Pre-processing of the dataset. (c) Training the proposed ALL-Net model to detect ALL. (d) Using the LIME algorithm to interpret the result output by ALL-Net. The entire process can be visualized in Fig. 3.

### Data collection

For this study, we considered the dataset consisting of 3,256 PBS images from 89 suspected ALL patients from Kaggle (*Aria et al., 2021*). The Taleqani Hospital's bone marrow laboratory produced the images for this dataset (Tehran, Iran). This dataset comprises two classes benign and malignant. The distribution of images can be seen in the Table 2. The

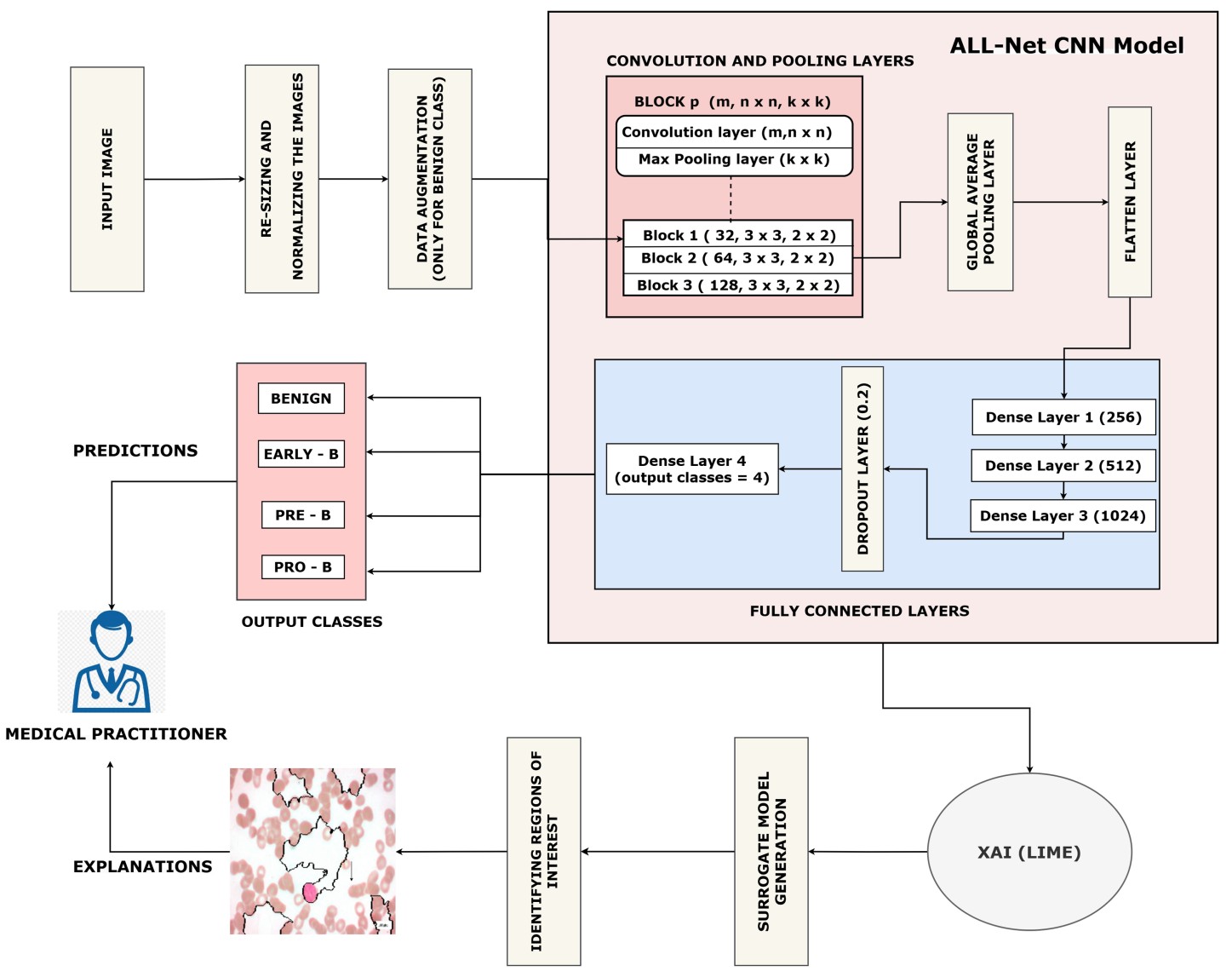

**Figure 3 Flow of the proposed ALL-Net model with data augmentation.**

**Table 2 Distribution of PBS images.**

| Type | Subtypes | Number of images |
|---|---|---|
| Benign | Hematogones | 504 |
| Malignant (ALL) | Early-B | 985 |
| | Pre-B | 963 |
| | Pro-B | 804 |
| Total | | 3,256 |

benign class consists of hematogones PBS images which are quite similar to ALL. The malignant class consists of PBS images of ALL subtypes: Early-B, Pre-B, and Pro-B ALL. The DOI for the dataset repository is 10.34740/KAGGLE/DSV/2175623.

**Table 3 Data augmentation parameters applied on the PBS image dataset.**

| Operation | Value |
| --- | --- |
| Rotation range | 10 |
| Horizontal flip | True |
| Height shift range | 0.1 |
| Shear range | 0.2 |
| Vertical flip | False |
| Width shift range | 0.1 |
| Fill mode | nearest |
| Zoom range | 0.2 |

## Preprocessing of dataset

All the images in the dataset are resized to dimensions of 224 × 224 pixels with three color channels (RGB), resulting in a shape of 224 × 224 × 3. This resizing ensures uniformity across the dataset, facilitating consistent input dimensions for the neural network. Additionally, the pixel values of each image are normalized to a range between 0 and 1. This normalization is achieved by dividing each pixel value by 255, as 255 is the maximum possible value for a pixel in an eight-bit image. The normalization process can be expressed mathematically by the Eq. (2). This normalization step is crucial as it helps to improve the convergence speed and stability of the neural network during training, ensuring that the model learns more efficiently by keeping the input data within a consistent and manageable range.

$$Normalized\ pixel\ value = \frac{Pixel\ value}{255}. \tag{2}$$

The distribution of blood cell images of each class can be observed in Table 2. From table, it is clear that there are fewer images in the benign class than in the other classes. This may create a data imbalance problem (*Singamsetty et al., 2024*) in which a model does not learn much about a class with fewer images compared to the other classes with more images. To overcome and tackle the data imbalance problem, images are added to the benign class using the data augmentation technique. For this dataset, we applied the image or data augmentation only on the benign class images with the parameters mentioned in Table 3. Before and after performing data augmentation, the distribution of images for each class can be seen in Fig. 4. All four classes now have images in a similar range. The proposed ALL-Net model is then processed on both augmented and unaugmented datasets in the following sections, and the resulting outcomes are examined in the results and discussions section.

## Proposed convolution neural network

Next, with an 80:20 ratio, the pre-processed image dataset is split into two subsets: the training set and the testing set. This choice guarantees the model access to a significant volume of data for parameter adjustment and learning, which could result in improved
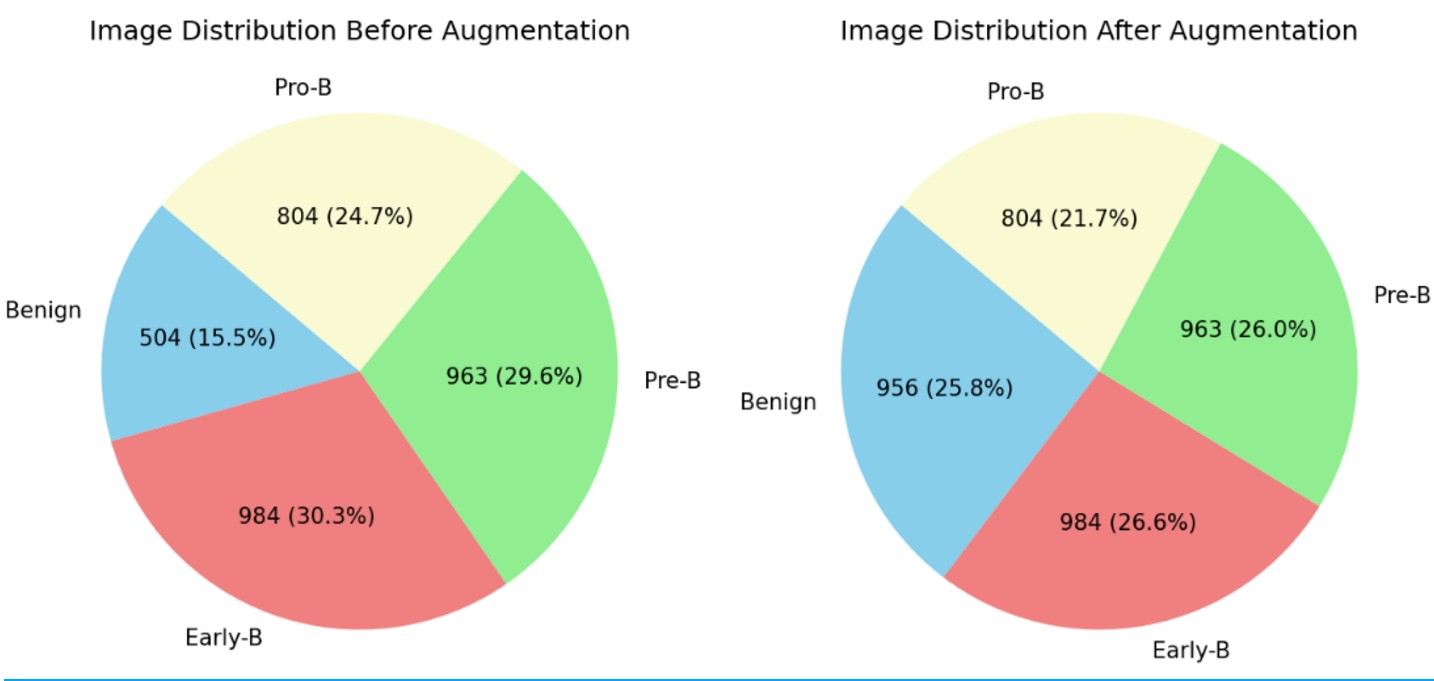

**Figure 4 Distribution of images before data augmentation.**

training and better performance on unknown data during testing. The entire proposed architecture of the proposed ALL-Net model can be visualized in Fig. 5. The ALL-Net model commences with an input layer accommodating pictures of dimensions 224 × 224 × 3. Three blocks are then routed across the input image. Each block comprises a max-pooling layer and a convolution layer with a ReLU activation (*Lundberg & Lee, 2017*) function. To lower spatial dimensions and computational complexity, the first block includes a max pooling layer with a size of 2 × 2 and a convolution layer with 64 filters and a 3 × 3 filter size. The second block consists of a max pooling layer with dimensions of 2 × 2 and a convolution layer with 128 filters of 3 × 3 size. The final component of the third block is the max pooling layer, which is 2 × 2, and the third convolution layer, which has 256 filters of size 3 × 3. A global average pooling layer is used after the blocks to combine spatial information from different feature maps and enable efficient feature representation. The 2D feature maps are then converted into a 1D vector using a flatten layer. To understand intricate patterns and relationships found in the data, the flattened feature vector is subsequently fed through a succession of densely or fully connected layers, each consisting of a particular number of neurons. The initial dense layer consists of 256 neurons followed by a ReLu layer. Then, 512 neurons in a second dense layer are used, and ReLU activation occurs once more. The final dense layer has 1,024 neurons. To avoid overfitting, a dropout layer is added with a rate of 0.2 after the three fully connected layers, randomly dropping a fraction of 20 percent of neurons during training. The network concludes with a dense layer comprising a number of neurons equal to the output classes, which in this case is four. The softmax activation function is utilized to compute the probability distribution across the

```
Model: "sequential"
```

| Layer (type) | Output Shape | Param # |
|---|---|---|
| conv2d (Conv2D) | (None, 222, 222, 32) | 896 |
| max_pooling2d (MaxPooling2D) | (None, 111, 111, 32) | 0 |
| conv2d_1 (Conv2D) | (None, 109, 109, 64) | 18,496 |
| max_pooling2d_1 (MaxPooling2D) | (None, 54, 54, 64) | 0 |
| conv2d_2 (Conv2D) | (None, 52, 52, 128) | 73,856 |
| max_pooling2d_2 (MaxPooling2D) | (None, 26, 26, 128) | 0 |
| global_average_pooling2d (GlobalAveragePooling2D) | (None, 128) | 0 |
| flatten (Flatten) | (None, 128) | 0 |
| dense (Dense) | (None, 256) | 33,024 |
| dense_1 (Dense) | (None, 512) | 131,584 |
| dense_2 (Dense) | (None, 1024) | 525,312 |
| dropout (Dropout) | (None, 1024) | 0 |
| dense_3 (Dense) | (None, 4) | 4,100 |

**Figure 5 Architecture of the proposed CNN.**

classes, with each neuron in this layer corresponding to one class. After designing the architecture, we optimized the model using the Adam optimizer, which facilitates efficient convergence. For the loss function, we employed sparse categorical cross-entropy, well-suited for multi-class classification. The sparse categorical cross-entropy loss is defined as:

$$Loss = -\frac{1}{N}\sum_{i=1}^{N} log(\hat{y}_{i,true}) \tag{3}$$

where $\hat{y}_{i,true}$ is the predicted probability of the true class for sample $i$.

Next, with a batch size of 32 and the number of epochs equal to 50, ALL-Net is trained on both augmented and unaugmented image datasets. Ultimately, the performance of the trained ALL-Net model is assessed using common evaluation measures, including precision, f1 score, recall, and accuracy on an independent test set.

The code for this program is written in Python with Jupyter as an Integrated Development Environment (IDE). The entire project is simulated on a DELL XPS 13 computer with the following specifications: Intel i5 processor, 8GB RAM, and 64-bit OS, x64-based processor.

## RESULTS AND DISCUSSION

In this section, we discuss the performance of our proposed ALL-Net model on the PBS image dataset and compare the results with some existing works. We used the following metrics to evaluate our model.

1. **Accuracy**: Accuracy is the ratio of correctly classified samples to the total number of samples. Its mathematical formula is:

$$Accuracy = \frac{TP + TN}{TP + TN + FP + FN}. \qquad (4)$$

2. **Precision**: Precision is the ratio of the number of true positive samples to all positive predictions made by the model. Its mathematical formula is:

$$Precision = \frac{TP}{TP + FP}. \qquad (5)$$

3. **Recall or sensitivity**: Recall is the ratio of true positive predictions to all actual positive samples in the dataset. Its mathematical formula is:

$$Recall = \frac{TP}{TP + FN}. \qquad (6)$$

4. **F1 score**: F1 score is the harmonic mean of precision and recall. It provides a balance between precision and recall. Its mathematical formula is:

$$F1 = \frac{2 \cdot Precision \cdot Recall}{Precision + Recall}. \qquad (7)$$

In the above equations:

- TP–True positives: The predicted class is the same as the actual class
- TN–True negatives: The predicted class is the same as the actual class
- FP–False positives: The model predicted positive even though the actual sample is negative
- FN–False negatives: The model predicted the result as negative even though the actual sample is positive

For both the unaugmented and augmented image datasets, the results of training and testing data on the ALL-Net model are detailed in Tables 4, and 5. We used recall, accuracy, precision, and F1-score to evaluate the model. For the unaugmented dataset, the testing accuracy is 97.85% and the training accuracy is 98.42%. Figures 6 and 7 illustrate the epochs *vs.* loss and epochs *vs.* accuracy graphs for this dataset, where it can be observed that the accuracy curve shows an upward trend, and the loss curve shows a downward trend. For the augmented image dataset, the testing accuracy improved to 99.32%, and the training accuracy increased to 99.59%. Figures 8 and 9 depict the epochs *vs.* loss and epochs *vs.* accuracy graphs for this dataset. Similar to the unaugmented dataset, the accuracy curve for the augmented dataset shows an upward trend, and the loss curve displays a downward trend. These trends for graphs of both datasets further confirm that our proposed model is fitting correctly, demonstrating robustness and effectiveness in

**Table 4 Training and testing data results for unaugmented image dataset.**

|  | Category | Precision | Recall | F1-Score | Accuracy |
|---|---|---|---|---|---|
| Training | Benign | 99% | 100% | 99% | 98.42% |
|  | Early | 100% | 99% | 100% |  |
|  | Pre | 100% | 100% | 100% |  |
|  | Pro | 100% | 100% | 100% |  |
| Testing | Benign | 93% | 97% | 95% | 97.85% |
|  | Early | 98% | 95% | 97% |  |
|  | Pre | 100% | 99% | 100% |  |
|  | Pro | 99% | 100% | 100% |  |

**Table 5 Training and testing data results for augmented image dataset.**

|  | Category | Precision | Recall | F1-Score | Accuracy |
|---|---|---|---|---|---|
| Training | Benign | 100% | 100% | 100% | 99.59% |
|  | Early | 100% | 100% | 100% |  |
|  | Pre | 100% | 99% | 100% |  |
|  | Pro | 99% | 100% | 99% |  |
| Testing | Benign | 100% | 100% | 100% | 99.32% |
|  | Early | 98% | 100% | 99% |  |
|  | Pre | 100% | 98% | 99% |  |
|  | Pro | 99% | 100% | 100% |  |

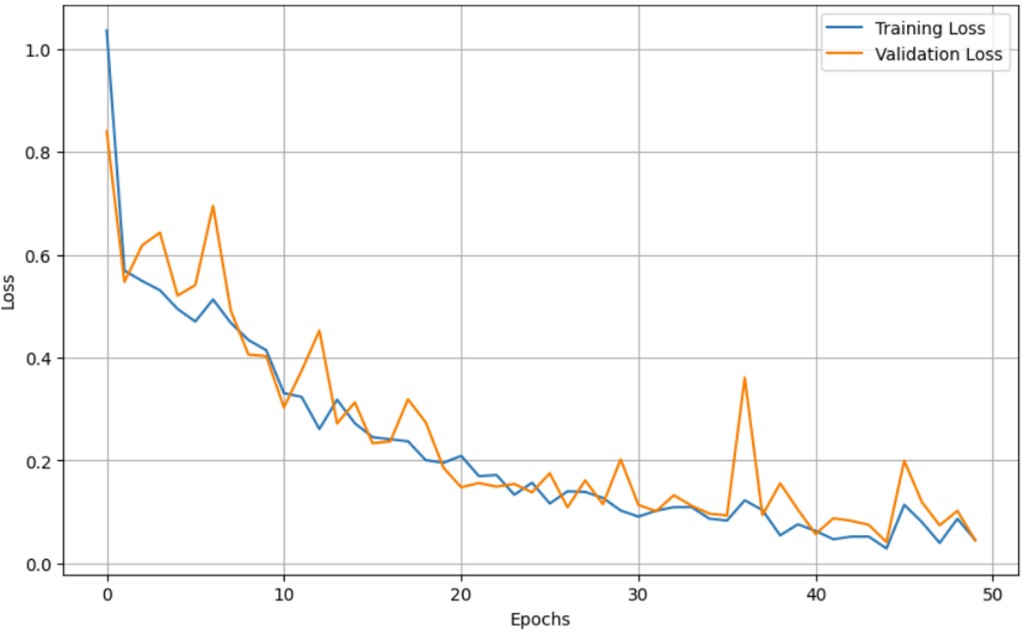

**Figure 6 Epochs *vs*. Loss graph for unaugmented data.**

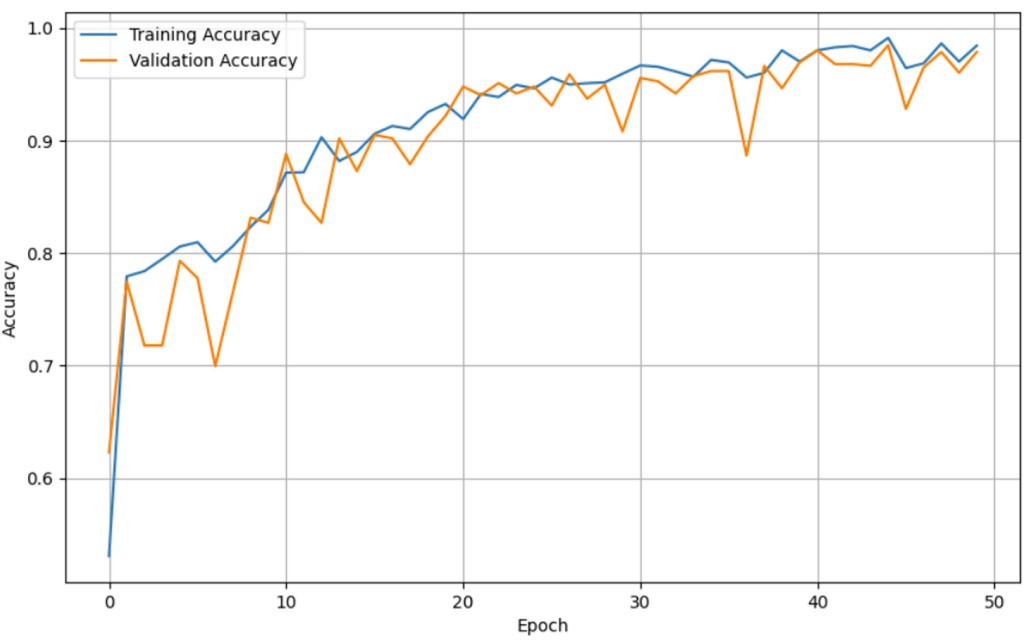

**Figure 7** Epoch *vs.* Accuracy graph for unaugmented data.

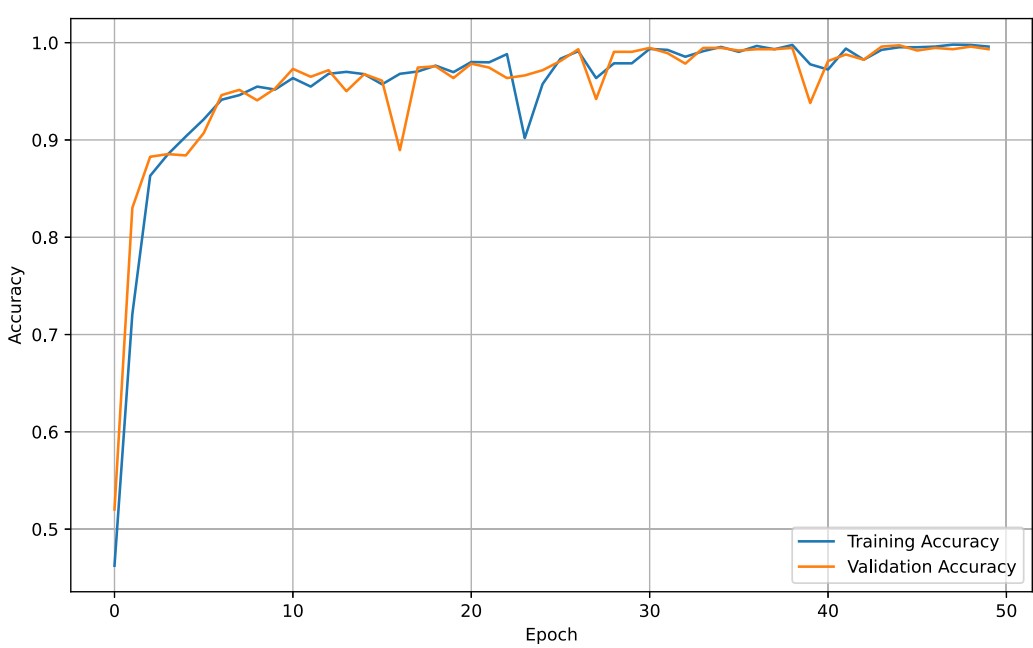

**Figure 8** Accuracy *vs.* Epoch graph for augmented data.

learning from both unaugmented and augmented image data. This comprehensive evaluation highlights the model's capability to generalize well across different types of datasets.

In addition to accuracy, the other metrics of the model's performance are as follows. For the unaugmented training dataset, the model achieved mean values of 99.75% for

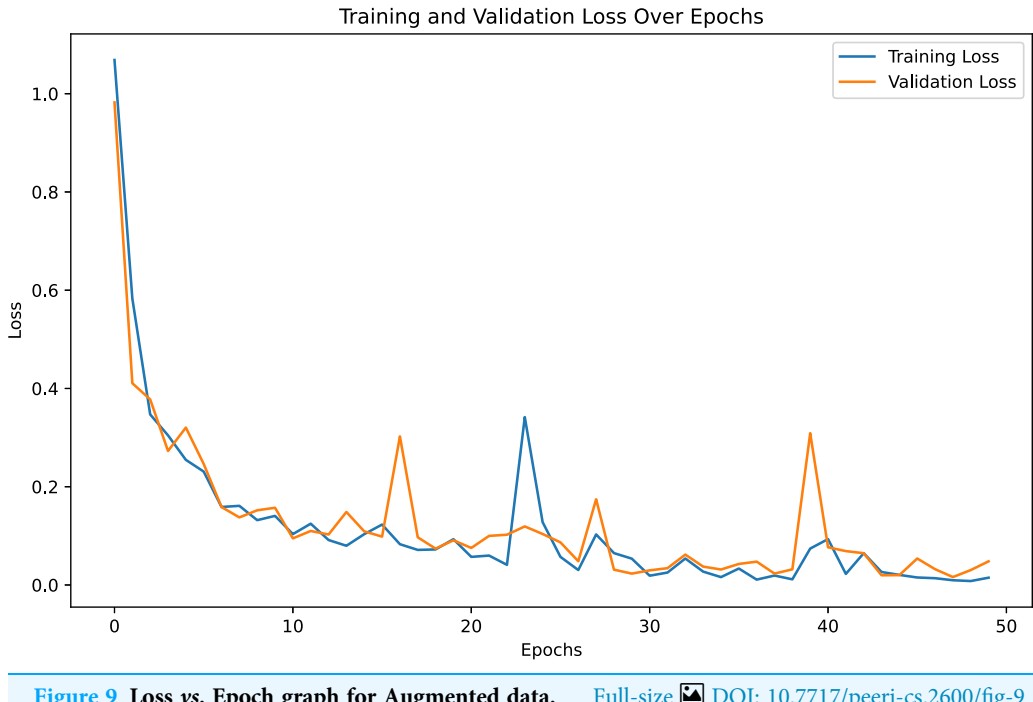

**Figure 9** **Loss** *vs.* **Epoch graph for Augmented data.**

precision, recall, and F1 score. On the unaugmented testing set, these metrics were 97.5%, 97.75%, and 98.0%, respectively. For the augmented training dataset, the precision, recall, and F1 score remained consistent at 99.75%. However, on the augmented test set, the model achieved a precision of 99.25%, with recall and F1 score both reaching 99.5%.

The confusion matrix in Fig. 10 for the ALL-Net model applied to the unaugmented dataset reveals that a significant number of images in the benign class are misclassified. This misclassification indicates that the model struggled to accurately distinguish benign images from other classes due to the class imbalance problem. However, after applying data augmentation techniques to the dataset, there is a notable improvement in the model's performance. The updated confusion matrix, shown in Fig. 11, demonstrates that the number of misclassified images in the benign class is reduced to zero. This reduction highlights the effectiveness of data augmentation in enhancing the model's ability to correctly classify benign images, thereby improving the overall accuracy and robustness of our customized CNN ALL-Net model.

Our proposed ALL-Net model, when fine-tuned on the target dataset with data augmentation, demonstrates superior performance compared to many pretrained models which are also fine-tuned on the same augmented dataset. Table 6 compares the testing data results of ALL-Net with various pretrained CNNs used as feature extractors by *Ghaderzadeh et al. (2022)*. Our ALL-Net outperformed all pretrained CNN models except for DenseNet201, which achieved a slightly higher accuracy of 99.85% compared to ALL-Net's 99.32%. Despite this slight difference in accuracy, ALL-Net is significantly less complex than DenseNet201, which comprises numerous layers. A reduced number of layers implies that our model can operate faster than those with more layers. The results

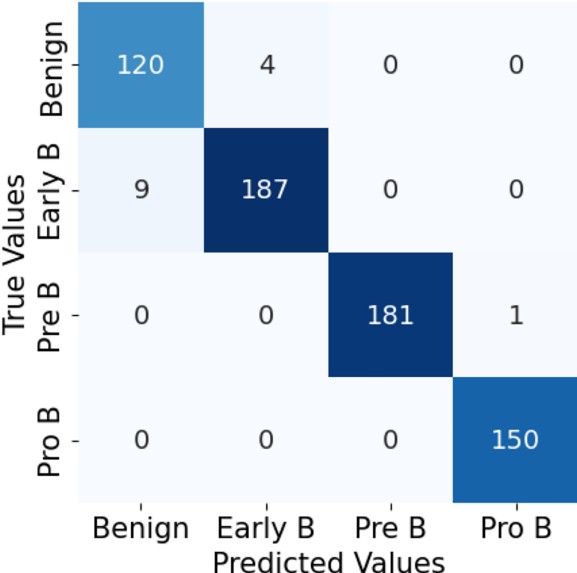

**Figure 10 Confusion matrix for CNN on unaugmented PBS images.**

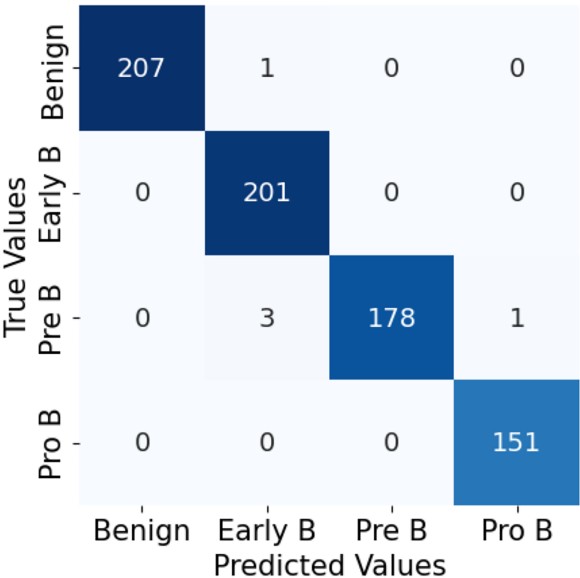

**Figure 11 Confusion matrix for CNN on augmented PBS images.**

align with our main objective: highlighting the need for a more customized and less complex model, such as ALL-Net, specifically designed for ALL classification.

For interpretation of the results generated by ALL-Net, we utilized the XAI technique LIME. It allows us to highlight the regions in each image that contribute most significantly to the model's classification decision for a given class. For instance, consider the image displayed in Fig. 12, which belongs to the "Pre" class. ALL-Net correctly classifies this image as "Pre". LIME is then applied by perturbing the input image 1,000 times (a default

**Table 6 A comparison of the testing accuracy of our proposed ALL-Net model *vs.* other pre-trained networks that are used as feature extractors on the augmented dataset.**

| Network used | Accuracy |
| --- | --- |
| EfficientNet | 28.22% |
| MobileNetV3 | 50.15% |
| VGG-19 | 96.32% |
| Xception | 96.70% |
| InceptionV3 | 96.93% |
| ResNet50V2 | 97.85% |
| VGG-16 | 98.01% |
| NASNetLarge | 98.16% |
| DenseNet201 (*Ghaderzadeh et al., 2022*) | 99.85% |
| Proposed ALL-Net | 99.32% |

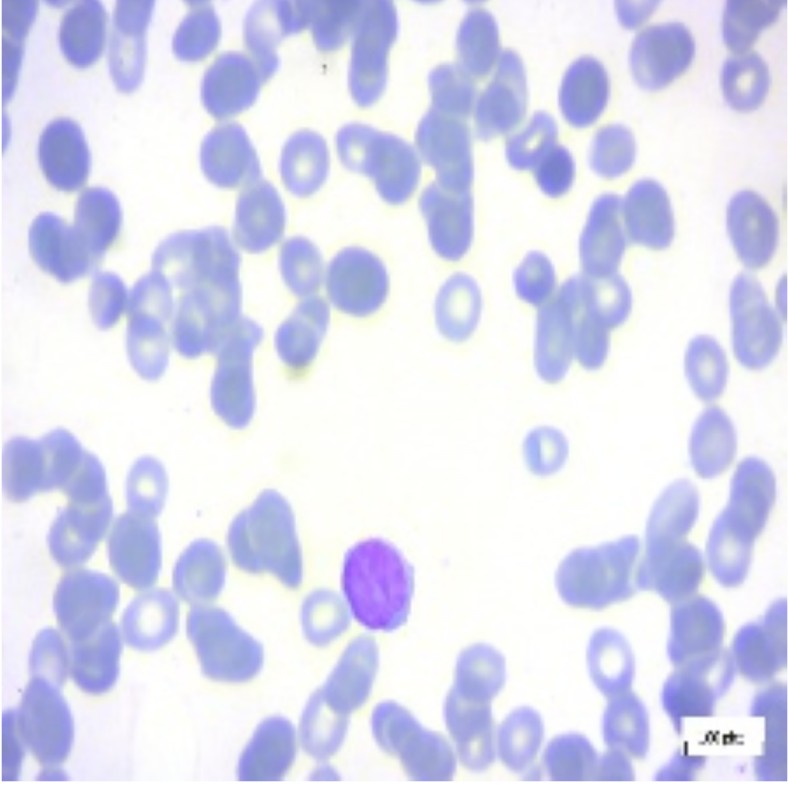

**Figure 12 Pre-B ALL image.**  

value) and observing how these perturbations affect the model's predictions. This process allows us to identify the areas that most influence the model's decision by generating the explanations. As shown in Fig. 13, we overlay a mask on the image to indicate the top five most influential regions for the ALL-Net model's classification. Figure 14 further isolates

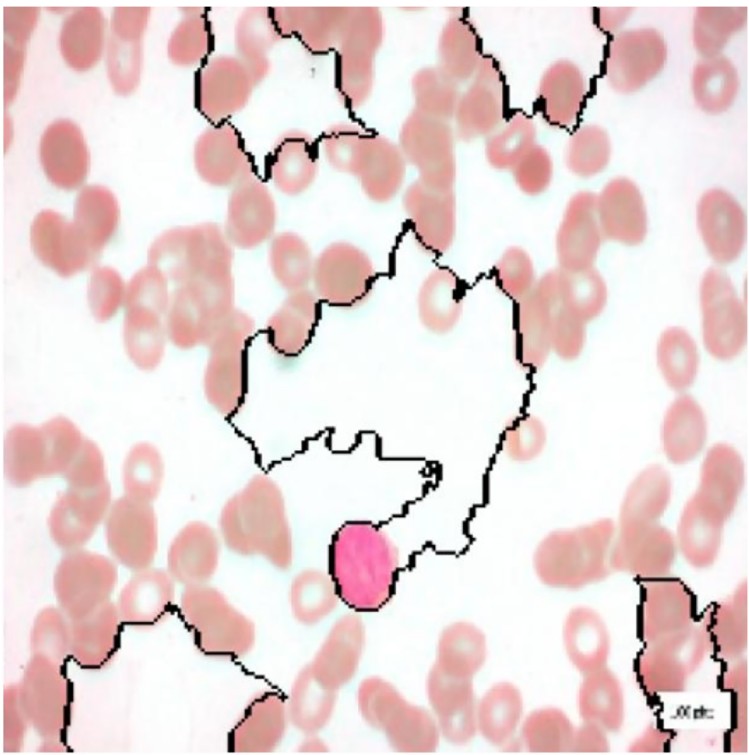

**Figure 13 Full image with regions contributing the most.**

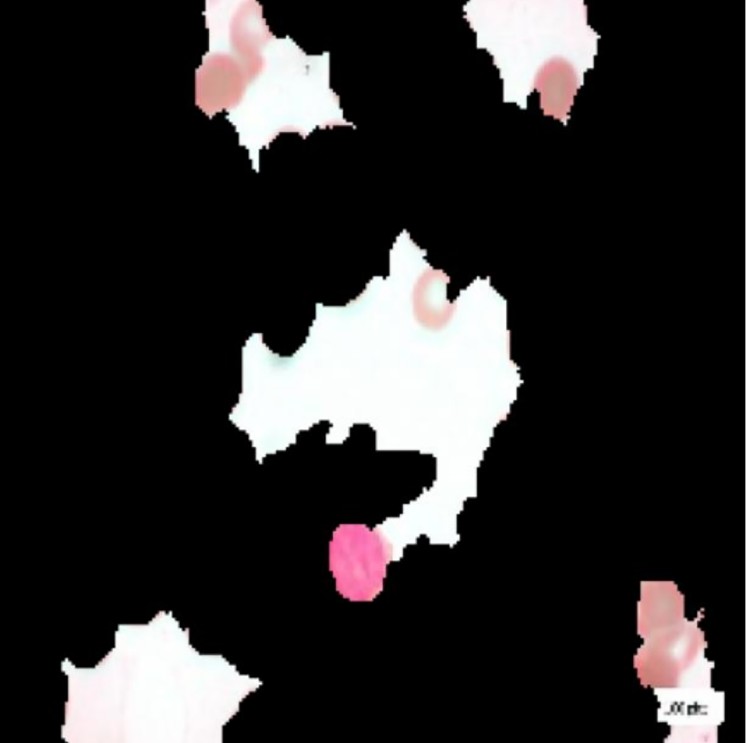

**Figure 14 Image with only the regions contributing the most.**

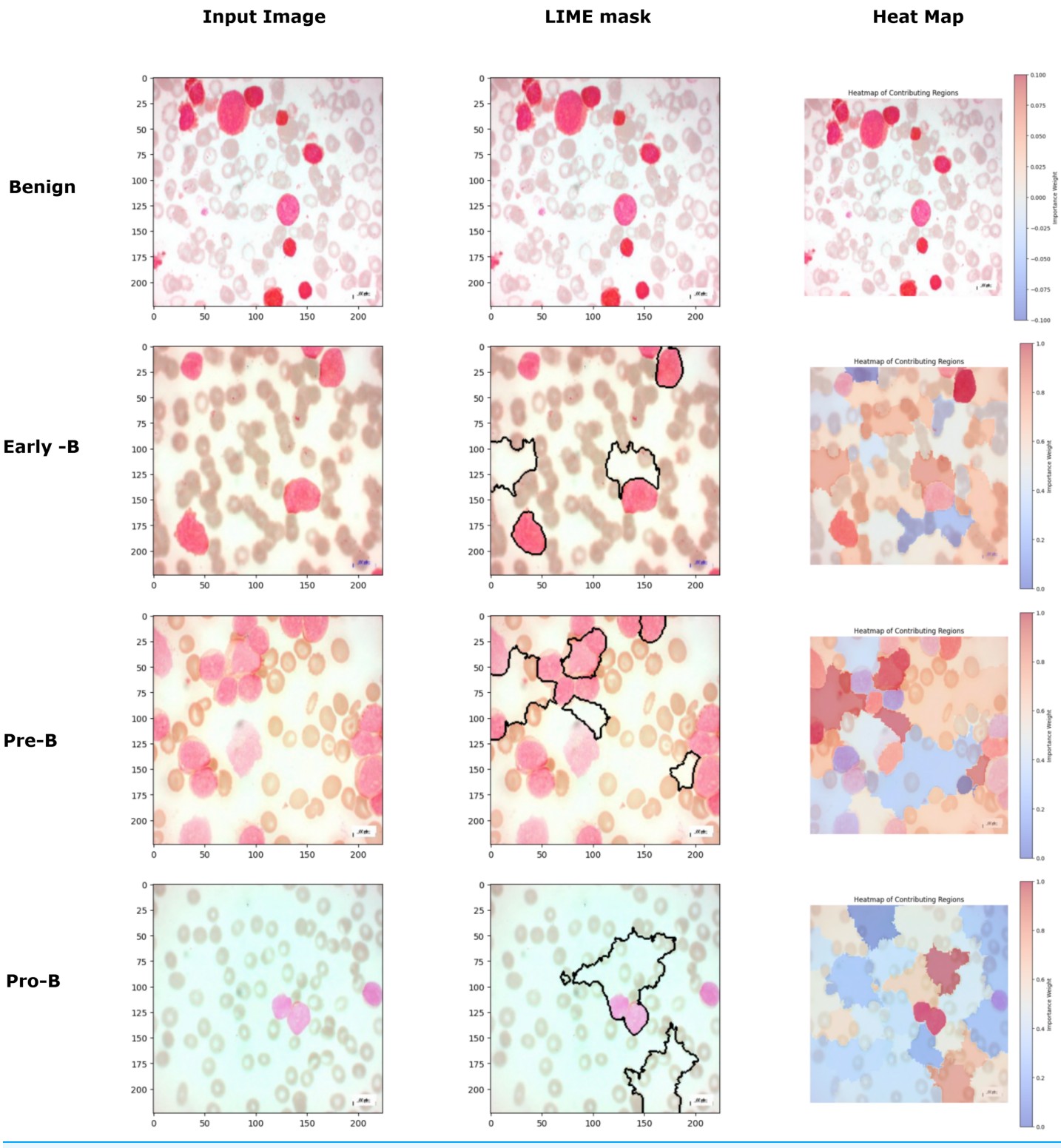

**Figure 15  Comparison of LIME mask and heatmaps on random PBS images.**     

these regions, focusing solely on the areas that were most critical for the model's classification.

Figure 15 image further demonstrates the application of the LIME to interpret model predictions for classifying different types of blood cells. Each row corresponds to a distinct class (Benign, Early-B, Pre-B, and Pro-B), showing three columns: the input image, the LIME mask, and a heatmap of contributing regions. The input image provides the original cell sample, while the LIME mask highlights the top five regions that contributed most to the model's classification, outlined in black. In the heatmap column, the contributing regions are color-coded to indicate their influence, with red representing areas that positively contribute to the model's classification and blue representing areas with negative contributions. This visualization helps reveal the specific areas the model focuses on for each classification, enabling a more interpretable understanding of how the model differentiates between the various cell types. For benign class images, though the model predicts the images correctly, the LIME mask and the heatmap are not generated for that class because there are no regions in the image to classify that the image belongs to a malignant or cancerous class. This approach provides valuable insight into the specific image features ALL-Net relies on to make its predictions, enhancing the interpretability and transparency of the model's decision-making process.

## CONCLUSION AND FUTURE WORK

To conclude, our work demonstrates the potential of ALL-Net in detecting ALL from PBS images. Using a dataset comprising 3,256 PBS images across four classes of Benign, Early, Pre and Pro, our ALL-Net model attained an accuracy of 97.85%. Additionally, we addressed the class imbalance challenge through data augmentation techniques, particularly the benign class. This showed a substantial improvement in accuracy, reaching 99.32%. Overall, this work shows the capability of deep learning in transforming medical diagnoses and enhancing patient care, and it adds to the expanding body of literature on CNN-based approaches for leukemia identification. We can take advantage of cutting-edge technology and creative approaches to further the area of medical image analysis and eventually improve patient outcomes globally.

Moving forward, further research could explore additional techniques to enhance model performance, such as fine-tuning model architectures, optimizing hyperparameters, and exploring advanced data augmentation strategies. Also, to make predictions more accurate and help the models learn, enormous volumes of data are required. However, there is currently limited medical data available. In addition, patients must feel certain that their information will not be altered or exploited before providing it. Therefore, adding a security feature to this model will be quite beneficial.

### Funding
The authors received no funding for this work.

## Competing Interests

The authors declare that they have no competing interests.

## Author Contributions

- Abhiram Thiriveedhi conceived and designed the experiments, performed the experiments, analyzed the data, performed the computation work, prepared figures and/or tables, authored or reviewed drafts of the article, and approved the final draft.
- Swetha Ghanta conceived and designed the experiments, performed the experiments, analyzed the data, performed the computation work, prepared figures and/or tables, authored or reviewed drafts of the article, and approved the final draft.
- Sujit Biswas conceived and designed the experiments, performed the experiments, analyzed the data, performed the computation work, authored or reviewed drafts of the article, and approved the final draft.
- Ashok K. Pradhan conceived and designed the experiments, performed the experiments, analyzed the data, performed the computation work, authored or reviewed drafts of the article, and approved the final draft.

## Data Availability

The data is available at Kaggle: 10.34740/kaggle/dsv/2175623.

The code is available at GitHub and Zenodo:

https://github.com/Abhiram014/ALL-Net-Detection-of-ALL-using-CNN-and-XAI

Thiriveedhi, A., Ghanta, S., & Pradhan, A. K. (2024). ALL Net-Detection of ALL using CNN and XAI. Zenodo. https://doi.org/10.5281/zenodo.14349780

## Supplemental Information

Supplemental information for this article can be found online at http://dx.doi.org/10.7717/peerj-cs.2600#supplemental-information.

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
