# Peer review of "ALL-Net: integrating CNN and explainable-AI for enhanced diagnosis and interpretation of acute lymphoblastic leukemia"

_PeerJ Computer Science, doi:10.7717/peerj-cs.2600_

## Round 0.1 · original submission · Major Revisions

Dear authors,

Your article has not been recommended for publication in its current form. However, we do encourage you to address the concerns and criticisms of the reviewers and resubmit your article once you have updated it accordingly.

Best wishes,

·

Basic reporting

Manuscript ID Submission ID 103511v1
This paper is related to reviewing the manuscript titled " ALL-Net:Integrating CNN and Explainable-AI for enhanced diagnosis and interpretation of Acute Lymphoblastic Leukemia"
The new ALL-Net model for the precise diagnosis of acute lymphoblastic leukemia (ALL) is presented in this publication. Using a unique Convolutional Neural Network (CNN) architecture combined with Explainable Artificial Intelligence (XAI) approaches, ALL-Net successfully classifies pictures of peripheral blood smears (PBSs) into three ALL subtypes (Early B, Pre-B, and Pro-B) and one benign category (hematogones). Three thousand six hundred PBS pictures make up the dataset used to train and assess the algorithm.
Firstly, Although the proposed study is successful in terms of organization, presentation, content and results, major revision given in the following items need to be performed.

Experimental design

Provide the major numerical findings and conclusions of the study in the abstract section another than accuracy performance criterion.
2) Use abbreviations after the first use in the text, in the abstract and throughout the paper, and for example XAI (Explainable AI) is used twice.
3) There seems to be no need to use the title "Leukemia" in the introduction.
4) The standard CNN architecture is depicted in Fig. 5. I think the graphical scheme of the proposed CNN-XAI method should be given.
5) The title "DATA AUGMENTATION" is used twice in the article. It should be checked.
6) Instead of the general explanation of Explainable AI (XAI), its mathematical model should be given and how it is integrated with CNN should be verified with mathematics and equations.
7) While explaining the "Data augmentation" method in Fig.6, it would be more meaningful if leukemia images were shown instead of a dog image and exemplified in this way. For example, how can an Ealy-b diseased image be evaluated in different angular ways?
8) It would be better to include Fig.8 in the data collection and/or data augmentation section. Similarly, Fig. 10 should be placed in the methodology section, not in the results section.
9) In the results and performance analysis, classical accuracy and loss graphs were drawn and interpreted. Why was the loss function not defined? Performance results should be increased by other important metrics. In addition, more professional code designs such as heatmap should be used and graphs should be obtained for the representation of the confusion matrix. In this sense, figs 15 and 16 seem very simple and do not comply with the quality of the journal.
10) The segmentation results in Fig. 18 and 19 need to be verified numerically. How can we understand how accurate this segmentation is?

Validity of the findings

As above

Additional comments

My decision is major revision. I do not see any harm in publishing the manuscript once the above revisions are made.
Best regards.

Reviewer 2 ·

Basic reporting

Overall, the quality of the paper presentation is acceptable, but there are several areas that could be improved, particularly in writing and figure explanations. For example, in line 52, the phrase "There are several ways ALL (can)" is unclear—what is meant by this? Additionally, starting from line 268, the following sentence is redundant, as the second sentence does not provide any new information:
"Next, with an 80:20 ratio, the pre-processed image dataset is split into two subsets: the training set and the testing set. Eighty percent (more than half) of the dataset is used for model training. This choice guarantees the model access to a significant volume of data for parameter adjustment and learning, which could result in improved training and better performance on unknown data during testing"

In terms of figure explanations, for example, in Figures 1-4, although examples are provided, the lack of detailed explanation limits the reader’s understanding of the expected differences between the various data classes. As someone not specialized in the field, I noticed that the purple cell's color density seems to be a significant difference between the classes. If this is the case, I question the necessity of employing complex and powerful deep neural networks for classification.

Experimental design

The experimental design is insufficient for publication for the following reasons:

1. The authors claim that the proposed method qualifies as explainable AI (xAI). However, this is difficult to justify with the use of stacked convolutional and fully connected layers. The visualizations in Figures 17-19 provide only limited insight into region-wise contributions. To substantiate this claim, the authors should: (1) demonstrate that other models cannot produce visualizations aligned with human intuition, and (2) present a full heatmap of the image, rather than focusing on curated regions with missing information in other areas, as is currently the case.

2. The benchmarking against other models is not comprehensive. If the other models being compared were only pretrained on general datasets like ImageNet, it is expected that they would underperform compared to the proposed method, which was fine-tuned directly on the target dataset. Additionally, when the proposed model is fine-tuned without augmentation, its performance falls below that of classical models like VGG and ResNet. This does not effectively validate the proposed model's performance.

Validity of the findings

Overall, I believe the current draft fails to validate the effectiveness of the proposed method. Detailed information regarding these concerns can be found in the experimental design section of my review.

---

## Round 0.2 · accepted · Accept

Dear Authors,

Thank you for the revised paper. One of the reviewers of previous round declined to assess the latest revision. However, another reviewer has accepted your paper. In addition, I believe that your paper has been sufficiently improved and is now ready for publication following the second revision.

Best wishes,

·

Basic reporting

The authors have completed all the suggested and requested corrections and revisions in the first round. Therefore, I find it appropriate to publish the article after its final review and control so that it can move on to the printing process of the Journal.

Experimental design

None

Validity of the findings

None